# Influence of 2-Nitroimidazoles in the Response of FaDu Cells to Ionizing Radiation and Hypoxia/Reoxygenation Stress

**DOI:** 10.3390/antiox12020389

**Published:** 2023-02-06

**Authors:** Faisal Bin Rashed, Wisdom Deebeke Kate, Mesfin Fanta, Leonard Irving Wiebe, Piyush Kumar, Michael Weinfeld

**Affiliations:** Department of Oncology, University of Alberta, Edmonton, AB T6G 2R3, Canada

**Keywords:** hypoxia, nitroimidazole, head and neck cancer, radiosensitizer, DNA damage

## Abstract

Cellular adaptations to hypoxia promote resistance to ionizing radiation (IR). This presents a challenge for treatment of head and neck cancer (HNC) that relies heavily on radiotherapy. Standard radiosensitizers often fail to reach diffusion-restricted hypoxic cells, whereas nitroimidazoles (NIs) [such as iodoazomycin arabinofuranoside (IAZA) and fluoroazomycin arabinofuranoside (FAZA)] can preferentially accumulate in hypoxic tumours. Here, we explored if the hypoxia-selective uptake of IAZA and FAZA could be harnessed to make HNC cells (FaDu) susceptible to radiation therapy. Cellular response to treatment was assessed through clonogenic survival assays and by monitoring DNA damage (immunofluorescence staining of DNA damage markers, γ-H2AX and p-53BP1, and by alkaline comet assay). The effects of reoxygenation were studied using the following assays: estimation of nucleoside incorporation to assess DNA synthesis rates, immunofluorescent imaging of chromatin-associated replication protein A as a marker of replication stress, and quantification of reactive oxygen species (ROS). Both IAZA and FAZA sensitized hypoxic HNC cells to IR, albeit the former is a better radiosensitizer. Radiosensitization by these compounds was restricted only to hypoxic cells, with no visible effects under normoxia. IAZA and FAZA impaired cellular adaptation to reoxygenation; high levels of ROS, reduced DNA synthesis capacity, and signs of replication stress were observed in reoxygenated cells. Overall, our data highlight the therapeutic potentials of IAZA and FAZA for targeting hypoxic HNC cells and provide rationale for future preclinical studies.

## 1. Introduction

Solid tumours often display an imbalance between O_2_ consumption and supply, mostly due to unrestrained cell growth and vascular deformities. This creates regions in the tumour mass where cells are sub-optimally oxygenated [1]. Hypoxic tumours resist treatment, such as radiation therapy (RT), which primarily depends on the availability of molecular O_2_ to generate cytotoxic effects [2]. This is particularly problematic for cancers, such as locally advanced or unresectable tumours of the head and neck, where radiation is the mainstay of treatment [3]. Indeed, 5-year survival for head and neck cancer (HNC) patients receiving standard treatment is only about 40–50%, with locoregional recurrence occurring in up to 30% of patients [4]. Importantly, hypoxia has been identified as a major driver for therapy resistance and poor outcome in HNC patients [5,6]. Therefore, exploring approaches to sensitize hypoxic tumour cells to radiotherapy has the potential to greatly improve therapeutic outcome in HNC patients.

A promising strategy to target therapy-resistant hypoxic tumours is using hypoxia-activated prodrugs (HAPs; such as nitroimidazoles, NIs). These agents become selectively activated and accumulate only in hypoxic cells by bioreductive binding to the cellular macromolecules [7]. NIs preferentially affect hypoxic cell viability and often display hypoxia-selective radiosensitization potential in cell and animal models [8,9]. They have shown particular benefit in patients with head and neck tumours [10]; for example, addition of nimorazole (a 5-NI) to RT can improve treatment outcome in HNC patients with hypoxic tumours [11]. The present study is focused on two clinically validated 2-NI hypoxic radiotracers, iodoazomycin arabinofuranoside (IAZA) and fluoroazomycin arabinofuranoside (FAZA) (Figure 1A,B) [12,13,14]. Previously, we have shown that these compounds can selectively target hypoxic cells and induce cytostasis [15]. Additionally, their capacity to sensitize hypoxic human hepatocellular carcinoma and colorectal cancer cells to ionizing radiation (IR) has been reported elsewhere [16,17]. However, if and how IAZA and FAZA affect response of hypoxic HNC cells to IR is yet to be explored in detail.

In addition to low O_2_ stress, solid tumours are often subjected to reoxygenation due to irregular blood circulation [18,19]. Termed cyclic hypoxia, this dynamic change in oxygenation can further complicate cancer treatment. While conventional understanding dictates that reoxygenation likely makes cells susceptible to RT, recent findings paint a more ambiguous picture. In fact, reoxygenation-mediated oxidative stress has been linked to stabilization of hypoxia inducible factor 1 alpha (HIF-1α) transcription factor, which can lead to resistance to RT [20]. Moreover, introduction of O_2_ following chronic hypoxia can activate additional molecular pathways, such as Hedgehog and lysyl oxidase, which ultimately lead to manifestation of a more aggressive disease [21,22,23]. Interestingly, effects of NIs on cellular response to reoxygenation have rarely been investigated.

Bearing these observations in mind, this study aimed to assess the implications of combining IAZA or FAZA with IR in HNC cells and whether and how they influence cellular response to hypoxia/reoxygenation stress. The present manuscript should be considered a companion piece to our previous work, and together, they provide a detailed picture of the therapeutic potentials of these compounds in targeting hypoxic tumours.

## 2. Materials and Methods

Cell culture: Experiments were carried out with a human head and neck squamous cell carcinoma cell line, FaDu, which was purchased directly from American Type Culture Collection (Manassas, VA, USA). Cells were cultured in DMEM/F-12 medium supplemented with 10% fetal bovine serum, 1% penicillin streptomycin, and 1% 2 mM L-glutamine, and routinely checked for mycoplasma contamination. Normoxic experiments were carried out in a humidified incubator at 37 °C with 5% CO_2_. To induce hypoxia, we used either an in-house degassing/regassing system fitted with metal canisters (<0.1% O_2_) [24] or a humidified chamber equipped with controlled O_2_ flow (0.1% O_2_) (ProOx P110, BioSpherix, Parish, NY, USA). For hypoxic experiments, cells were grown in glass petri dishes.

Drugs and treatment conditions: IAZA [25] and FAZA [26] were synthesized as described elsewhere. Stock solutions of IAZA (500 mM), FAZA (500 mM), and hydroxyurea (HU; 1 M, H8627, Sigma-Aldrich, Oakville, ON, Canada) were prepared in DMSO. N-Acetyl-L-cysteine (NAC; A7250, Sigma-Aldrich, Oakville, ON, Canada) was dissolved in a 1:1 ratio of DMSO and distilled H_2_O, and used at a final concentration of 3 mM. Treatment with IAZA and FAZA lasted for 24 h under respective O_2_ conditions; 0.02% DMSO was used as a vehicle control. For radiation response assays, cells were irradiated (dose rate 0.5 Gy/min) with a ^60^Co source Gammacell irradiator (AECL, Chalk River, ON, Canada) 1 h prior to the end of the drug incubation. For recovery/reoxygenation studies, drug-containing medium were replaced with fresh medium (containing 0.02% DMSO), and cells were processed as indicated after 24 h of recovery. For reactive oxygen species (ROS) experiments, treatment lasted for 72 h (20% or 0.1% O_2_), followed by a 24-h recovery period.

Clonogenic survival assay: Clonogenic assays were performed using glass petri dishes. Drug (IAZA and FAZA) and radiation treatment were carried out as described earlier. Following treatment, cells were then allowed to grow and form colonies in drug-free medium for 14 days under normoxia, after which colonies were stained with crystal violet, counted manually, and plotted as percentage of vehicle-treated non-irradiated controls. For sensitizer enhancement ratio (SER) calculations, colony counts were plotted as percentage of respective non-irradiated vehicle or drug-treated controls. Sensitizer enhancement ratio (SER) values were calculated with GraphPad Prism V7 (GraphPad Software, La Jolla, CA, USA) using the Fit Spline function (smoothening spline, with number of knots set at 4) at 10% surviving fraction for each treatment condition.

Immunocytochemistry: FaDu cells grown on sterilized glass coverslips were treated as indicated, followed by fixation in 2% paraformaldehyde (PFA, P6148, Sigma-Aldrich, Oakville, ON, Canada). Fixed cells were blocked in 1% BSA and probed with anti-phospho-histone H2A.X (Ser139) (γ-H2AX; 1:5000 dilution, 05-636, Millipore Sigma, Burlington, MA, USA, or 1:1000 dilution, 39118, Active Motif, Carlsbad, CA, USA), anti-phospho-53BP1 (Ser1778) (1:2000 dilution, PA5-17462, Invitrogen, Waltham, MA, USA), or anti-replication protein A2 (RPA; 1:2500 dilution, ab2175, Abcam, Cambridge, UK). Secondary antibodies used include Alexa Fluor 488 conjugated anti-mouse IgG and Alexa Fluor 594 anti-rabbit IgG (1:1000 dilution; Molecular Probes, Eugene, OR, USA). For RPA staining, cells underwent a nuclear extraction phase (25 mM HEPES, 300 mM sucrose, 50 mM NaCl, 1 mM EDTA, 3 mM MgCl_2_, 0.5% NP-40, pH 7.9) for 5 min prior to fixation in PFA. Hoechst 33342 (Life Technologies, Carlsbad, CA, USA) was used to counterstain nuclei. Mounted coverslips on glass slides were imaged with a Plan-Apochromat 40X/1.3 Oil DIC lens on a Zeiss 710 confocal microscope using Zen 2011 software.

Alkaline comet assay: FaDu cells treated with IAZA and FAZA were allowed to recover for 24 h and processed for alkaline comet assay as described earlier [15]. Ethidium bromide (E7637, Sigma-Aldrich, Oakville, ON, Canada) stained comet slides were imaged with a digital upright 2 microscope (ZEISS AxioImager.Z1, Carl Zeiss, Jena, Germany). The CometScore™ software (TriTek Corp., Sumerduck, VA, USA) was used to analyze comets.

Click-iT EdU incorporation assay: To assess the effects of IAZA and FAZA on DNA synthesis following reoxygenation, FaDu cells treated with drugs (100 µM) or vehicle control (0.02% DMSO) were allowed to recover for 24 h in drug-free medium containing 10 µM 5-ethynyl-2′-deoxyuridine (EdU, C10339, Invitrogen, Waltham, MA, USA). PFA-fixed cells were processed for “click” staining following the manufacturer’s protocol (C10339, Invitrogen) and imaged on a Zeiss 710 confocal microscope.

Immunoblotting: Cell lysates prepared in RIPA buffer were used for immunoblots. Primary antibodies used include: anti-hypoxia inducible factor 1 alpha (1:2000 dilution; NB100-449, Novus Biologicals, Littleton, CO, USA), anti-beta actin (1:2000 dilution; sc-47778, Santa Cruz Biotechnology, Dallas, TX, USA), anti-protein phosphatase 2A (1:2000 dilution; 05-421, Millipore Sigma) and anti-beta tubulin (1:4000 dilution; ab6046, Abcam). Membranes were probed with the following secondary antibodies: IR-800 conjugated goat anti-mouse IgG (1:2000 dilution, 926-80010, LI-COR, Lincoln, NE, USA) and goat anti-rabbit-IgG-HRP (1:2000 dilution; Jackson Immunoresearch, West Grove, PA, USA). Membrane scans were obtained with an Odyssey Fc imager (LI-COR).

Assessing hydrogen peroxide (H_2_O_2_) levels: Levels of H_2_O_2_ were used as a readout of reactive oxygen species (ROS) and were analyzed using a ROS-Glo™ H_2_O_2_ assay kit (G8820, Promega, Madison, WI, USA). Briefly, FaDu cells seeded in 96-well plates were treated as indicated for 72 h, followed by 24 h of recovery in drug-free medium containing either 0 or 3 mM NAC. Afterwards, cells were incubated with the H_2_O_2_ substrate for 4 h, which generates a luciferin precursor upon reaction with H_2_O_2_. Finally, 50 µL of medium from each well was used to measure the luciferin luminescence with an OPTIMA microplate reader (BMG Labtech, Ortenberg, Germany). Cells in the original plate were assayed for viability with crystal violet as described previously [15]. Luminescence data were normalized by the cell viability results.

Data processing and statistics: Micrographs and immunoblots were adjusted for brightness and contrast using Adobe Photoshop (Adobe Inc., San Jose, CA, USA). Image Studio lite v5.2 (LI-COR) was used to quantify immunoblots, which were then normalized to vehicle-treated normoxia controls. γ-H2AX and RPA staining intensity within the nucleus was quantified with IMARIS software (Bitplane, Zürich, Switzerland) using Hoechst staining as a nuclear mask. For γ-H2AX data analysis, intensity values were divided into arbitrary intervals, which were then plotted on the x-axis, and the percent population within that intensity range was plotted on the y-axis. A maximum threshold was chosen so that the intensity of ~95% of cells in the vehicle-treated control fell below this threshold, and the percentage of cells above this threshold were considered positive for γ-H2AX. Graph preparation and statistical analysis were performed with GraphPad Prism V7. The 2-tailed unpaired t-test was used for statistical analysis, with *p* < 0.05 considered statistically significant. Asterisks depict statistically significant differences: ns (not significant), * (*p* ≤ 0.05), ** (*p* ≤ 0.01), *** (*p* < 0.001), and **** (*p* < 0.0001). Graphs display the mean with standard error of the mean (S.E.M.).

## 3. Results

IAZA and FAZA sensitize hypoxic HNC cells to ionizing radiation. Colony formation assays were performed to test cellular response to IAZA and FAZA (100 µM), alone or in combination with ionizing radiation (IR, 5 Gy). Stabilization of HIF-1α was used as an indicator to confirm successful induction of hypoxia (Figure 1C). As reported earlier, hypoxic cells displayed preferential sensitivity towards 2-NI treatment [15]. Hypoxic incubation with IAZA or FAZA (100 µM, 24 h) decreased colony counts by ~50% and ~25%, respectively; no effects were seen under normoxia (Figure 1D). As anticipated, hypoxic cells were >2 times more resistant to IR than their normoxic counterparts. Combining IAZA or FAZA with radiation reversed this radioresistant phenotype. Upon irradiation, IAZA-treated hypoxic cells showed a significant reduction in clonogenicity (~17% survival) when compared to vehicle-treated hypoxic cells (~50% survival); indeed, combining IAZA treatment with RT reduced survival of the hypoxic cells to the same extent as the normoxic irradiated cells (~20% survival). Hypoxic cellular response to FAZA+ IR was less drastic (~37% survival). To further confirm that these effects are not a cell-line-specific phenomenon, some of these findings were validated in a glioblastoma model (A172), where we observed a similar pattern of sensitization by FAZA (Appendix A).

The radiosensitization effects by IAZA and FAZA at 5 Gy, however, appeared to be mostly additive. At higher radiation dosage, IAZA and IR showed a synergistic response with an SER value of 1.41 (Figure 1E). FAZA treatment (100 µM) under hypoxia generated a more modest SER value of 1.09 (Figure 1F). Importantly, no radiosensitization was seen under normoxia with either IAZA or FAZA, demonstrating their safety to healthy cells.

IAZA and FAZA increase DNA damage in hypoxic irradiated cells. To test if co-treatment of hypoxic cells with the drugs and IR increases DNA damage, we performed immunofluorescent staining for phosphorylated histone 2A (γ-H2AX) as a readout of DNA double-strand breaks. Irradiated hypoxic cells (5 Gy) showed a significant decrease in γ-H2AX-positive population when compared to their normoxic counterparts, confirming the resistance of hypoxic cells to IR (Figure 2A,B). Interestingly, IAZA treatment under hypoxia alone generated an increase in γ-H2AX-positive population similar to that seen in the hypoxic group treated with radiation only (~30%). Furthermore, combined treatment with IAZA and IR under hypoxia increased the γ-H2AX-positive population to ~60%. Hypoxic cells treated with FAZA alone and in combination with IR consisted of ~17% and 56% γ-H2AX-positive populations, respectively (Figure 2C). These observations were further validated by single-cell gel electrophoresis (alkaline comet assay), which directly measures the structural integrity of cellular DNA in the form of single-strand breaks and alkaline-labile sites. Radiation-induced DNA damage was significantly lower under hypoxia than normoxia, which was evident from a 50% reduction in comet tail moment seen in the hypoxic radiation treatment group (Figure 2D). In agreement with the γ-H2AX data, IAZA alone and in combination with IR showed a significant increase in comet tail moment, with the latter generating DNA damage levels similar to the irradiated normoxic cells. While FAZA alone under hypoxia did not generate any observable DNA damage in the comet assay, combination of FAZA with radiation under hypoxia also significantly increased DNA damage when compared to IR-treated hypoxic cells.

IAZA- and FAZA-treated hypoxic cells display high γ-H2AX staining following reoxygenation, but not a corresponding increase in DNA damage. To assess how IAZA- and FAZA-treated cells respond to exposure to oxygen after hypoxic incubation, drug-treated cells were allowed to reoxygenate for 24 h and stained for DNA damage markers, γ-H2AX, and phospho-53BP1 (a specific marker for DNA double-strand breaks). A marked increase in the percentage of cells positive for γ-H2AX was observed (Figure 3A,B). This effect was independent of radiation and only seen in drug-treated hypoxic cells that underwent reoxygenation. Qualitatively, these cells displayed a bright γ-H2AX signal, with a combination of punctate foci and pan-nuclear staining. Interestingly, phospho-53BP1 staining did not yield a similar response (Figure 3A), suggesting that the γ-H2AX signal was not exclusively due to DNA damage. Further confirmation for this finding came from the alkaline comet assay, where no change in comet tail moment was observed in the reoxygenated cells, regardless of the treatment conditions (Figure 3C). One possible explanation for the persistent γ-H2AX signal could be that treatment with IAZA and FAZA under hypoxia impaired cellular capacity to dephosphorylate γ-H2AX by reducing the cellular level of protein phosphatase 2A (PP2A), the enzyme primarily responsible for removing phosphorylation from H2A.X [27]. However, while hypoxia resulted in an overall reduction of total PP2A levels, no difference was observed among the various treatment combinations in the hypoxia group, regardless of whether cells were processed without any reoxygenation or after 24 h of reoxygenation (Figure 3D,E).

Hypoxic cells treated with IAZA and FAZA experience replication stress following reoxygenation. Since reoxygenated cells displayed a strong γ-H2AX signal but minimal DNA damage or reduced PP2A level, we wondered if these cells were experiencing anomalies in DNA synthesis, as replication stress can also lead to a pan-nuclear γ-H2AX signal [28]. To investigate this, drug-treated hypoxic cells were allowed to recover for 24 h in medium containing EdU, a nucleoside analogue that becomes incorporated into cellular DNA during replication. Overall, reoxygenated cells had lower EdU incorporation (Figure 4A,B), but it was significantly lower in drug-treated cells, suggesting IAZA- and FAZA-treated cells had impaired DNA replication capacities during reoxygenation. To further corroborate these observations, drug-treated reoxygenated cells were stained for replication protein A, which binds to single-stranded DNA generated as an intermediate from stalled replication forks [29]. Chromatin-bound RPA fraction is resistant to detergent extraction and, therefore, is indicative of the presence of single-stranded DNA and, by extension, of replication stress [30]. A significant increase in RPA-positive cells was observed in drug-treated cells in comparison to vehicle-treated cells upon reoxygenation; hydroxyurea-treated cells (24 h) served as positive control (Figure 4C,D).

To identify why these cells were experiencing replication stress, we decided to monitor cellular ROS levels, since reoxygenation is a major source of ROS [31], which can induce replication stress [32]. Drug-treated hypoxic cells, when allowed to reoxygenate for 24 h, showed a significant increase in ROS levels. The presence of a ROS scavenger (NAC) during the reoxygenation step led to a global reduction in cellular ROS levels (Appendix A). NAC incubation also significantly reduced levels of γ-H2AX (Figure 5A–D) and chromatin-bound RPA (Figure 5A,E) in drug-treated hypoxic/reoxygenated cells. This implies that higher levels of ROS from reoxygenation contributed, at least in part, to the generation of replication stress in drug-treated hypoxic/reoxygenated cells, which in turn manifested in the phosphorylation of γ-H2AX and enhanced chromatin retention of RPA.

## 4. Discussion

IAZA and FAZA are 2-NI-based hypoxia-activated molecules that have been clinically validated as radiotracers in different human malignancies, including head and neck cancers [12,33]. Our group has previously shown that in addition to their diagnostic utility, both compounds selectively prevent proliferation of hypoxic HNC cells (FaDu), and IAZA reduced tumour hypoxia levels in subcutaneous mouse HNC tumours [15]. However, whether these compounds can also make hypoxic HNC cells sensitive to RT was not known. To address this question, clonogenic survival assays were performed with FaDu cells treated with drug and radiation under normoxia and hypoxia (<0.1% O_2_). A relatively low concentration of each drug (100 µM) was used for our assays because (a) this would likely be reflective of clinically achievable dose, and (b) it would allow us to avoid concentration ranges that might induce clinical neurotoxicity, as reported with other NIs [34].

Combining IAZA or FAZA with 5 Gy IR under hypoxia significantly reduced colony formation when compared to vehicle-treated hypoxic irradiated cells (Figure 1D). In fact, hypoxic cells co-treated with IAZA and IR (5 Gy) showed similar clonogenicity as irradiated normoxic cells. Radiosensitization of hypoxic cells by IAZA and FAZA at 5 Gy IR generated mostly additive benefits, as the combined effects of drug and radiation was not greater than the cumulative effects of individual treatments. In fact, synergy only became apparent when drug-treated hypoxic cells were exposed to higher IR dosage: ≥12 Gy for IAZA, and ≥18 Gy for FAZA (Figure 1E,F). The SER value for IAZA (1.41) indicates that it is a better radiosensitizer than FAZA (SER 1.09) at the concentration tested. More importantly, the SER value for IAZA is comparable to, and perhaps better than, other NI hypoxic radiosensitizers (for example, glycididazole 1.29 [9], doronidazole 1.24 [9], misonidazole 1.2 [35] etc.).

Induction of DNA damage is widely used as a readout for efficacy of RT treatment. IR is ~2–3 times less efficient at inducing DNA damage under hypoxia than normoxia [36]. Our data agree with these observations and show that hypoxic FaDu cells respond poorly to IR, with overall lower levels of DNA damage from radiation exposure (Figure 2A–D). Using a surrogate marker for DNA double-strand breaks (γ-H2AX), we found that the percentage of cells containing IR-induced DNA damage under hypoxia is almost one-third of that seen under normoxia for the same radiation dose (Figure 2B,C). Further validation came from the direct assessment of DNA damage (using the alkaline comet assay) that showed almost 50% reduction in comet tail moment under hypoxia when compared to its normoxic counterpart (Figure 2D). In both assays, addition of IAZA or FAZA to IR under hypoxia significantly increased DNA damage, suggesting these compounds can potentiate the damaging effects of IR under hypoxia. Together with the SER analysis, our observations strongly support the need for future in vivo hypoxic radiosensitization studies with IAZA and FAZA.

The dynamic oxygenation status of solid tumours implies that cells must continuously adapt to low O_2_ as well as reperfusion cycles. While it is widely known that reoxygenation can lead to oxidative injury [37], how 2-NI compounds affect cellular response to reperfusion is poorly understood. To address this gap, we assessed two different surrogate DNA damage markers (γ-H2AX and phospho-53BP1) and directly measured DNA strand integrity using the alkaline comet assay in drug-treated hypoxic cells that were allowed to reoxygenate and recover for 24 h. Interestingly, drug-treated reoxygenated cells displayed very strong γ-H2AX levels, but not a corresponding increase in phospho-53BP1 staining (Figure 3A,B). Furthermore, no increase in DNA damage was observed in comet analysis (Figure 3C). This suggested that the increase in γ-H2AX levels was not primarily due to increased DNA damage. Cells also did not show any difference in total PP2A levels (Figure 3D,E), which implied these cells were not defective in their capacity to dephosphorylate γ-H2AX. We therefore turned our attention to replication stress, since it is a known inducer of H2AX phosphorylation [38]. Indeed, drug-treated hypoxic cells had reduced capacity for DNA replication, as observed by the EdU incorporation assay (Figure 4A,B), as well as higher levels of chromatin-bound RPA (Figure 4C,D), confirming that IAZA/FAZA-treated cells were undergoing replication stress. To find a contributor to this phenotype, cellular ROS levels were assayed, which were significantly higher in cells that were incubated with IAZA or FAZA under hypoxia and subsequently reoxygenated (Appendix A). Furthermore, when reoxygenation was carried out in the presence of NAC (a ROS scavenger), cellular levels of γ-H2AX and chromatin-bound RPA decreased significantly (Figure 5B–D). This suggests that IAZA and FAZA treatment under hypoxia makes cells susceptible to reoxygenation-mediated oxidative injury and induces replication stress. Nonetheless, it is worth noting that while NAC incubation during the reoxygenation phase led to a global (and significant) decrease in ROS levels (Appendix A), the relative levels of ROS between vehicle- and drug-treated reoxygenated cells remained relatively unchanged (Appendix A). This implies that other cellular factors might play a role in the observed phenotype.

## 5. Conclusions

In conclusion, a low dose of IAZA and FAZA can sensitize hypoxic HNC (FaDu) cells to IR. At the same concentrations, IAZA is a better hypoxic radiosensitizer than FAZA. Both compounds can disrupt cellular adaptation capacities to reoxygenation and lead to replication stress. These findings highlight the potential of IAZA and FAZA to be used in a therapeutic setting for the management of solid hypoxic head and neck tumours, and they position these compounds as dual function theranostic candidates, i.e., as therapeutic as well as diagnostic agents.

## Figures and Tables

**Figure 1 antioxidants-12-00389-f001:**
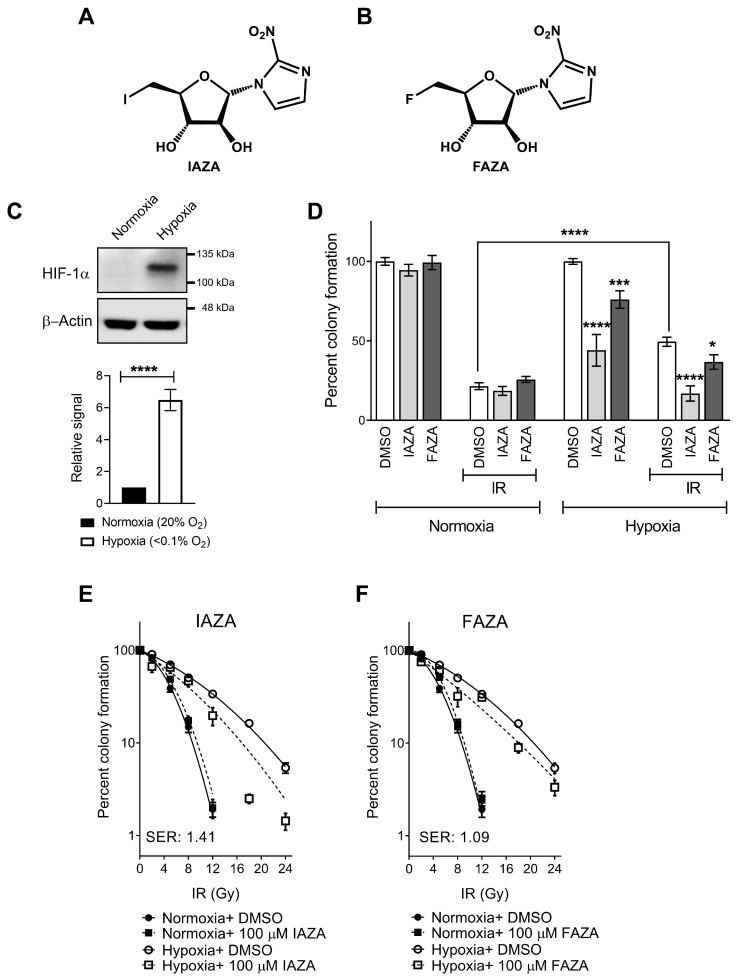
IAZA and FAZA sensitize hypoxic FaDu cells to IR. (**A**,**B**) Chemical structures for IAZA and FAZA. (**C**) Successful induction of hypoxia was confirmed by probing for HIF-1α. (**D**) IAZA and FAZA significantly reduced clonogenicity in hypoxic irradiated cells. (**E**,**F**) Sensitization by IAZA and FAZA appear to be synergistic at higher radiation dosage. The calculated sensitizer enhancement ratios (SER) for IAZA and FAZA are indicated. Data show mean from at least three independent experiments; error bars represent standard error of the mean (S.E.M.).

**Figure 2 antioxidants-12-00389-f002:**
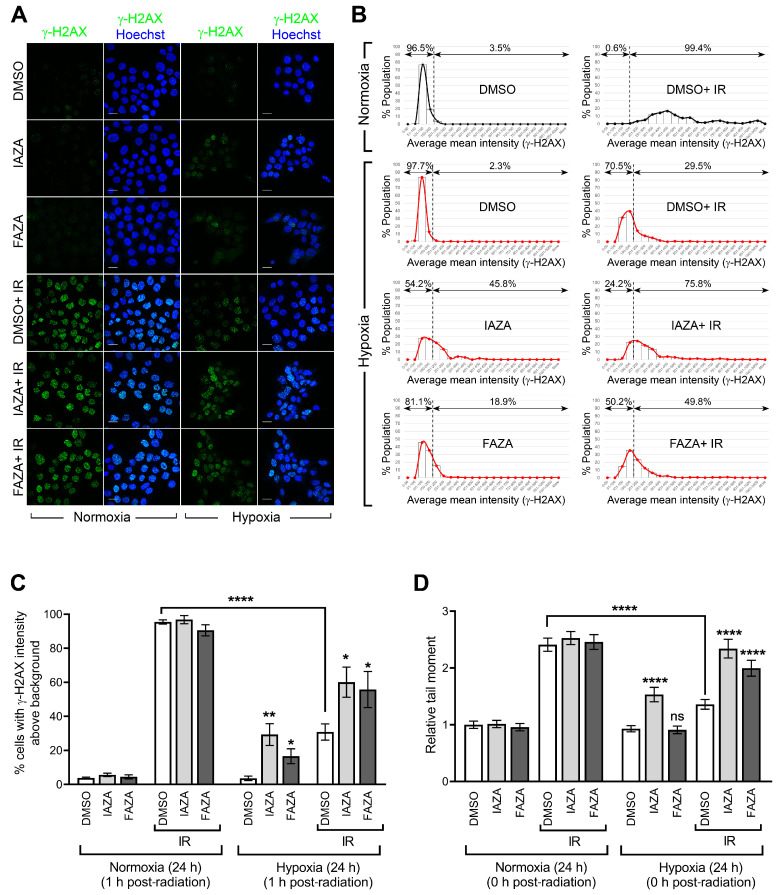
Co-treatment of IAZA or FAZA with 5 Gy IR increases DNA damage only under hypoxia. (**A**) Representative micrographs showing immunofluorescent staining of γ-H2AX in FaDu cells treated with vehicle control (DMSO) or drugs (IAZA or FAZA; 100 µM), with or without IR (5 Gy), under normoxia and hypoxia (<0.1% O_2_); scale bar = 20 µm. (**B**) Representative histograms showing quantification of γ-H2AX staining intensity; dotted line shows the threshold for background levels of γ-H2AX staining. (**C**) Treatment with IAZA and FAZA increases the percentage of γ-H2AX-positive population in hypoxic irradiated samples. (**D**) Alkaline comet assay shows significant increase in DNA damage in hypoxic drug-treated irradiated cells. Data show mean ± S.E.M. (**C**,**D**). The no-radiation arms of these data were previously reported in ref. [15] and are used here for comparison purposes.

**Figure 3 antioxidants-12-00389-f003:**
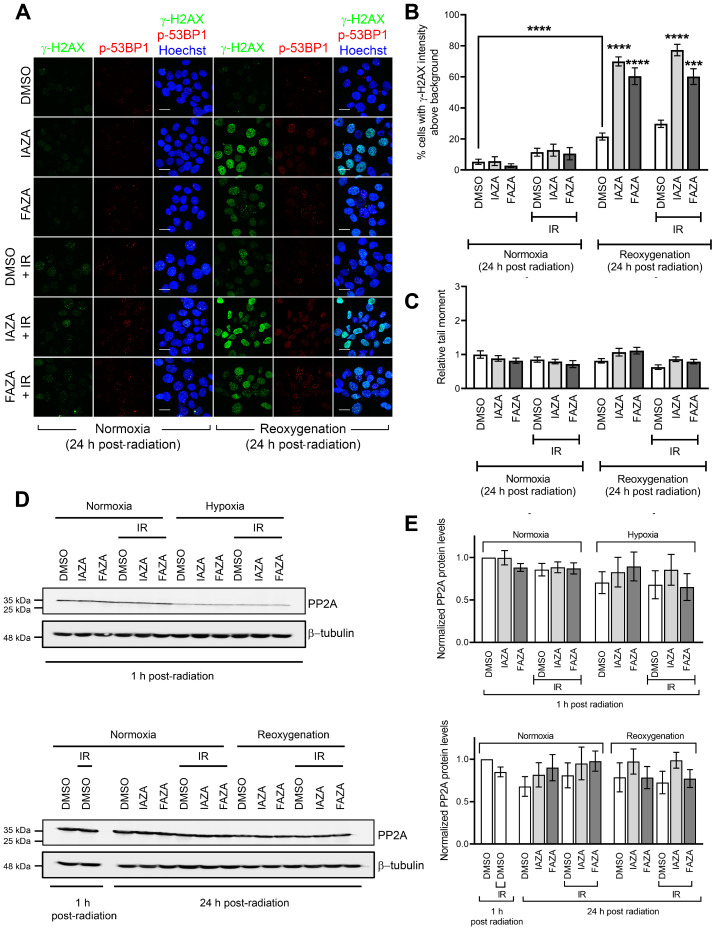
IAZA-/FAZA-treated hypoxic cells display high levels of γ-H2AX after reoxygenation, but not a corresponding increase in DNA damage. (**A**) Representative micrographs showing immunofluorescent staining of γ-H2AX and phopho-53BP1 in FaDu cells treated with vehicle control (DMSO) or drugs (IAZA or FAZA; 100 µM), with or without IR (5 Gy), under normoxia and hypoxia (<0.1% O_2_), followed by a 24 h reoxygenation and recovery period; scale bar = 20 µm. (**B**) Drug-treated reoxygenated cells showed increased percentage of γ-H2AX-positive population, but no increase in comet tail moment (**C**). (**D**) No effects were seen on total PP2A protein levels in hypoxia/reoxygenation group when cells were treated with IAZA or FAZA. (**E**) Quantification of PP2A immunoblots. Data show mean ± S.E.M from three independent replicates.

**Figure 4 antioxidants-12-00389-f004:**
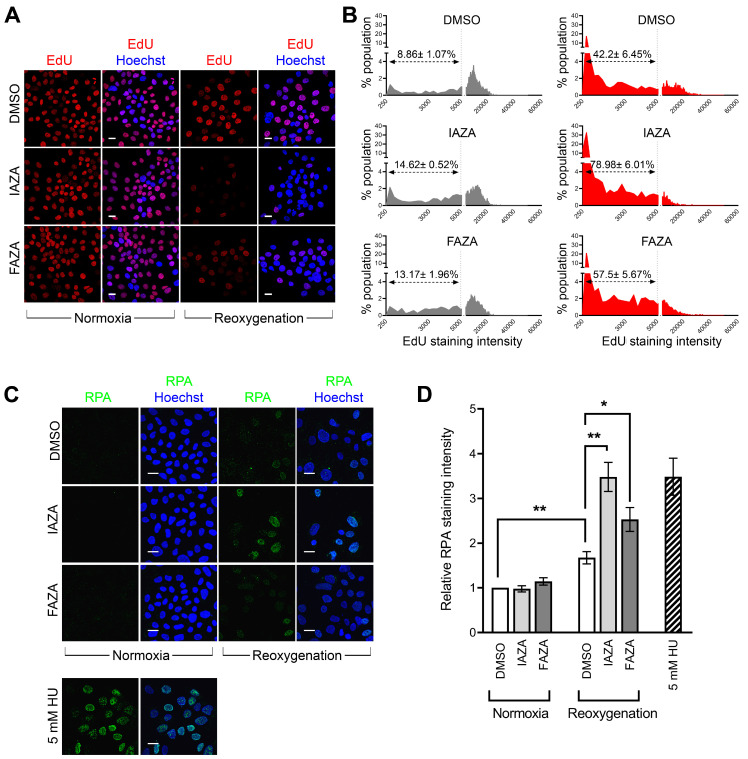
IAZA-/FAZA-treated hypoxic cells display signs of replication stress upon reoxygenation. (**A**) Representative micrographs showing click-EdU incorporation rates in FaDu cells treated with vehicle control (DMSO) or drugs (IAZA or FAZA; 100 µM) under normoxia and hypoxia (<0.1% O_2_), followed by a 24 h reoxygenation and recovery period. (**B**) Intensity of EdU click-stained micrographs was quantified and plotted as intensity versus %population histograms. Hypoxic exposure by itself increased cell population with low EdU staining; incubation with IAZA and FAZA under hypoxia further increased “low EdU stained” cell fractions. (**C**) Drug-treated reoxygenated cells showed higher levels of detergent extraction resistant chromatin-bound RPA. (**D**) Quantification of nuclear RPA staining. Data show mean ± S.E.M. from three independent replicates; scale bar = 20 µm.

**Figure 5 antioxidants-12-00389-f005:**
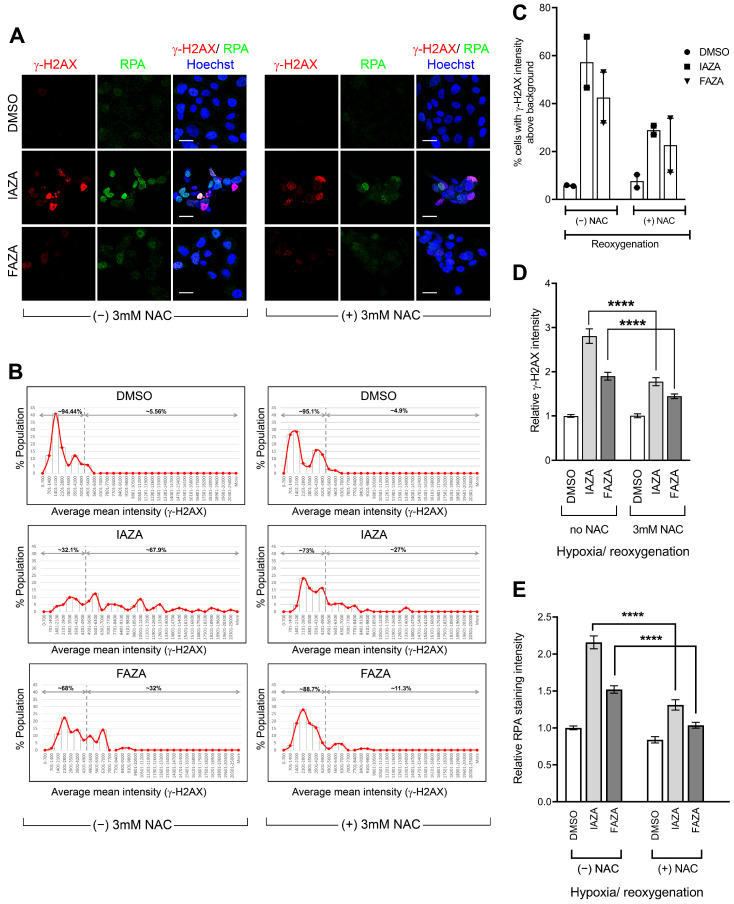
Reoxygenation in the presence of NAC reduced γ-H2AX and RPA-positive populations in drug-treated hypoxic cells. (**A**) Representative micrographs showing a reduction in γ-H2AX and RPA staining in response to NAC incubation during reoxygenation; scale bar = 20 µm. (**B**) Representative histograms showing quantification of γ-H2AX staining intensity; dotted line shows the threshold for background levels of γ-H2AX staining. (**C**) Percentage of γ-H2AX-positive population in hypoxia/reoxygenated cells treated with IAZA or FAZA (100 μM) was decreased in the NAC-treated group; data show mean ± S.E.M. from two independent replicates. Quantification of γ-H2AX (**D**) and RPA (**E**) staining intensity are shown; a minimum of 125 cells were analyzed for each condition.

## Data Availability

The data presented in this study are openly available in FigShare at https://figshare.com/s/4e180e079fb74370b486 (accessed on 28 December 2022).

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
