# Peer review of "Influence of 2-Nitroimidazoles in the Response of FaDu Cells to Ionizing Radiation and Hypoxia/Reoxygenation Stress"

_antioxidants, 2023, doi:10.3390/antiox12020389_

Round 1

Author Response

In manuscript “Influence of 2-nitroimidazoles in Cellular Response to Ionizing Radiation and Hypoxia/Reoxygenation Stress by Faisal Bin Rashed, Leonard Irving Wiebe, Piyush Kumar and Michael Weinfeld, authors studied cellular susceptibility for radiotherapy after drugs treatment. Manuscript is well written and very pleasant to read. Figures are clear and full of information. Manuscript can be considered for publication in Antioxidants after minor and rather technical corrections:

  • Lines283-285- please add reference.

We have now added a reference (line 285, reference 30).

  • Result section: The only difficulty with reading the text and fallowing the figures I found in this section: figures are large and text is separated in sub-sections concerning experiments, but text and figures are not always are located next to each other – authors should consider rearrangement of figures and text i.e. figure 5 is inside discussion section, but it also concerns figure 3: should be placed after the subsection 3.3.

We thank the reviewer for the suggestion and now have rearranged the figures accordingly.

  • Line:357 – not “several” rather “two different”.

We have updated the text as suggested (line 361).

Reviewer 2 Report

When looking for similar topics on Hypoxia – Cancer – Nitroimidazole, the following recently published material has been retrieved with Pubmed:

 Title : Cellular mechanism of action of 2-nitroimidazoles as hypoxia-selective therapeutic agents
Authors : Faisal Bin Rashed Diana Diaz-DussanFatemeh Mashayekhi Dawn Macdonald Patrick Nicholas NationXiao-Hong Yang Sargun Sokhi Alexandru Cezar Stoica Hassan El-SaidiCarolynne RicardoRavin Narain Ismail Hassan IsmailLeonard Irving WiebePiyush KumarMichael Weinfeld
Journal: Redox Biol 2022 Jun;52:102300.
 doi: 10.1016/j.redox.2022.102300. Epub 2022 Mar 21.

This paper is cited as reference 15 in the Antioxidants manuscript.

At least substantial part of the submitted material of the newly submitted manuscript in Antioxidants has already been published previously in Redox Biol. In the present paper, some pictures are obviously identical to the ones in Redox Biol, with a more detailed analysis (for example, the first four rows of figure 2A (containing 6 rows) in Redox Biol are identical to Figure 4 (containing 4 rows) in Antioxidants. The pictures were obviously obtained during the same unique experimental protocol.

The specific contribution in this paper for Antioxidants is to show that gamma-radiation toxicity is modified in cancerous cells exposed to hypoxia combined with nitroimidazoles. Apparently, some experimental data were already obtained during the same experiments conducted previously for Redox Biol.

In these conditions, this paper in Antioxidants has to be considered as a “complement” to Redox Biol article, and it should clearly be specified in each section of the manuscript. The reference 15 (which refers to the Redox Biol manuscript) should be more linked to parts of the paper which seems to duplicate same information.

In conclusion, the authors should clearly indicate the parts of the manuscript which duplicate previously published material. It can be understood that the authors reutilize some of their previously published data, but it must be explicit.

A rewriting to solve this question of “auto-plagiarism” seems required.

Author Response

When looking for similar topics on Hypoxia – Cancer – Nitroimidazole, the following recently published material has been retrieved with Pubmed:

Title : Cellular mechanism of action of 2-nitroimidazoles as hypoxia-selective therapeutic agents
Authors : Faisal Bin Rashed , Diana Diaz-Dussan, Fatemeh Mashayekhi , Dawn Macdonald , Patrick Nicholas Nation, Xiao-Hong Yang , Sargun Sokhi , Alexandru Cezar Stoica , Hassan El-Saidi, Carolynne Ricardo, Ravin Narain , Ismail Hassan Ismail, Leonard Irving Wiebe, Piyush Kumar, Michael Weinfeld
Journal: Redox Biol 2022 Jun;52:102300. 

doi: 10.1016/j.redox.2022.102300. Epub 2022 Mar 21.

This paper is cited as reference 15 in the Antioxidants manuscript.

At least substantial part of the submitted material of the newly submitted manuscript in Antioxidants has already been published previously in Redox Biol. In the present paper, some pictures are obviously identical to the ones in Redox Biol, with a more detailed analysis (for example, the first four rows of figure 2A (containing 6 rows) in Redox Biol are identical to Figure 4 (containing 4 rows) in Antioxidants. The pictures were obviously obtained during the same unique experimental protocol.

The specific contribution in this paper for Antioxidants is to show that gamma-radiation toxicity is modified in cancerous cells exposed to hypoxia combined with nitroimidazoles. Apparently, some experimental data were already obtained during the same experiments conducted previously for Redox Biol.

In these conditions, this paper in Antioxidants has to be considered as a “complement” to Redox Biol article, and it should clearly be specified in each section of the manuscript. The reference 15 (which refers to the Redox Biol manuscript) should be more linked to parts of the paper which seems to duplicate same information.

In conclusion, the authors should clearly indicate the parts of the manuscript which duplicate previously published material. It can be understood that the authors reutilize some of their previously published data, but it must be explicit.

A rewriting to solve this question of “auto-plagiarism” seems required.

We thank the reviewer for bringing up this point, We have now indicated in the figure legend for Fig. 2 that previous data published in reference 15 has been included for the sake of comparison. We also indicated in the Introduction the relationship between this paper and the Redox Biol paper (lines 66-68).

Reviewer 3 Report

In this manuscript, the authors investigated whether hypoxia-selective drugs IAZA and FAZA could make head and neck cancer cells susceptible to radiation. The authors demonstrated that IAZA and FAZA can induce radiosensitization against hypoxic head and neck cancer cell line FaDu. In addition, the authors suggested some of the mechanisms of their radiosensitization. I think that the topic and obtained results of this manuscript is interesting. However, in this manuscript, there are several flaws to be revised.

Comments:

1.      Only one cell line was used in this study. To validate the findings of this study, some of the main findings should be validated using other head and neck cancer cell lines.

2.      Dose rate is missing.

3.      To confirm hypoxic conditions, the protein expression of HIF-1alpha should be analyzed.

4.      Figure 5C: In line with Figure 2C and 3B, the percentages of gamma-H2AX positive cells should be shown. In addition, the information about the concentration of IAZA and FAZA is missing.

5.      Figure 5: The authors claimed that “ROS from reoxygenation contributed to the generation of replication stress in drug-treated hypoxic/reoxygenated cells” However, according to Figure 5A, C and D, the data of gamma-H2AX and RPA level in FAZA group without NAC is almost same that in IAZA group with NAC, whereas there is dramatic difference in ROS level among them. I think these data show replication stress in drug-treated hypoxic/reoxygenated cells is independent of ROS. In addition, the authors should clarify whether NAC recovers the NIs-decreased EdU incorporation and NIs-induced radiosensitization or not.

6.      Figure 5: I think that these data should be treated with multiple data. That is, the authors have to perform multiple comparison.

Author Response

In this manuscript, the authors investigated whether hypoxia-selective drugs IAZA and FAZA could make head and neck cancer cells susceptible to radiation. The authors demonstrated that IAZA and FAZA can induce radiosensitization against hypoxic head and neck cancer cell line FaDu. In addition, the authors suggested some of the mechanisms of their radiosensitization. I think that the topic and obtained results of this manuscript is interesting. However, in this manuscript, there are several flaws to be revised.

Comments:

  1. Only one cell line was used in this study. To validate the findings of this study, some of the main findings should be validated using other head and neck cancer cell lines.

We agree with the reviewer’s comment, but due to the short revision window, we could not source another head and neck cancer cell line. We have therefore modified the title to specifically state that the current manuscript deals with FaDu cells. We have also now added Supplementary Figure 1 showing that FAZA can act as a radiosensitizer in A172 glioblastoma cells.

  1. Dose rate is missing.

We now have mentioned the dose rate in the Materials and Methods section (line 85). 

  1. To confirm hypoxic conditions, the protein expression of HIF-1alpha should be analyzed.

We have now included HIF-1α data in Fig. 1C to confirm the successful induction of hypoxia.

  1. Figure 5C: In line with Figure 2C and 3B, the percentages of gamma-H2AX positive cells should be shown. In addition, the information about the concentration of IAZA and FAZA is missing.

We have reanalyzed our data according to the reviewer’s suggestions, and have included each in Figs. 5B and C. We have also indicated in the figure legend that the concentration of IAZA and FAZA was 100 microM.

  1. Figure 5: The authors claimed that “ROS from reoxygenation contributed to the generation of replication stress in drug-treated hypoxic/reoxygenated cells” However, according to Figure 5A, C and D, the data of gamma-H2AX and RPA level in FAZA group without NAC is almost same that in IAZA group with NAC, whereas there is dramatic difference in ROS level among them. I think these data show replication stress in drug-treated hypoxic/reoxygenated cells is independent of ROS. In addition, the authors should clarify whether NAC recovers the NIs-decreased EdU incorporation and NIs-induced radiosensitization or not.

And 6.      Figure 5: I think that these data should be treated with multiple data. That is, the authors have to perform multiple comparison.

In response to points 5 and 6, we have included Fig. S2 with the accompanying text in the Discussion “Nonetheless, it is worth noting that while NAC incubation during the reoxygenation phase led to a global (and significant) decrease in ROS levels (Fig. S2A), the relative levels of ROS between vehicle- and drug-treated reoxygenated cells remained relatively unchanged (Fig. S2B and C). This implies that other cellular factors might play a role in the observed phenotype.”

Round 2

Reviewer 2 Report

The authors took in consideration adequately the criticism of "auto-plagiarism". Their reference "15" citing the prior publication on a linked paper is repeated at adequate places within the manuscript.

Due to the complexity of the question, it is understandable that the authors presented their results in two linked papers, this one more specifically orientes to radiation toxicity on cancerous cells. 

Reviewer 3 Report

Thanks for responding to my comments.

I am almost satisfied with the authors' response.